# LATENT TOPIC CONVERSATIONAL MODELS

## ABSTRACT

Despite much success in many large-scale language tasks, sequence-to-sequence (seq2seq) models have not been an ideal choice for conversational modeling as they tend to generate generic and repetitive responses. In this paper, we propose a Latent Topic Conversational Model (LTCM) that augments the seq2seq model with a neural topic component to better model human-human conversations. The neural topic component encodes information from the source sentence to build a global "topic" distribution over words, which is then consulted by the seq2seq model to improve generation at each time step. The experimental results show that the proposed LTCM can generate more diverse and interesting responses by sampling from its learnt latent representations. In a subjective human evaluation, the judges also confirm that LTCM is the preferred option comparing to competitive baseline models.

## 1 INTRODUCTION

Sequence-to-Sequence model (seq2seq) (Sutskever et al., 2014), as a data-driven approach to mapping between two arbitrary length sequences, has attracted much attention and been widely applied to many natural language processing tasks such as machine translation (Cho et al., 2014; Luong et al., 2015), syntactic parsing (Vinyals et al., 2015), and summarisation (Nallapati et al., 2016). Neural conversational models (Vinyals & Le, 2015; Shang et al., 2015; Serban et al., 2016a) are the latest development in open-domain conversational modelling, where seq2seq-based models are employed for learning dialogue decisions in an end-to-end fashion. Despite promising results, the lack of explicit knowledge representations (or the inability to learn them from data) impedes the model from generating causal or even rational responses. This leads to many problems discussed in previous literature such as generic responses (Li et al., 2016a), inconsistency (Li et al., 2016b), and redundancy and contradiction (Shao et al., 2017).

On the other hand, goal-oriented dialogues (Young et al., 2013) use the notion of dialogue ontology to constrain the scope of conversation and facilitate rational system behaviour within the domain. Neural network-based task-oriented dialogue systems usually retrieve knowledge from a pre-defined database either by discrete accessing (Wen et al., 2017b; Bordes & Weston, 2017) or through an attention mechanism (Dhingra et al., 2017). The provision of this database offers a proxy for language grounding, which is crucial to guide the generation or selection of the system responses. As shown in Wen et al. 2017a, a stochastic neural dialogue model can generate diverse yet rational responses mainly because they are heavily driven by the knowledge the model is conditioned on.

Despite the need for explicit knowledge representations, building a general-purpose knowledge base and actually making use of it have been proven difficult (Matuszek et al., 2006; Miller et al., 2016). Therefore, progress has been made in conditioning the seq2seq model on coarse-grained knowledge representations, such as a fuzzily-matched retrieval result via attention (Ghazvininejad et al., 2017) or a set of pre-organised topic or scenario labels (Wang et al., 2017; Xing et al., 2016). In this work, we propose a hybrid of a seq2seq conversational model and a neural topic model – Latent Topic Conversational Model (LTCM) – to jointly learn the useful latent representations and the way to make use of them in a conversation. LTCM uses its underlying seq2seq model to capture the local dynamics of a sentence while extracts and represents its global semantics by a mixture of topic components like topic models (Blei et al., 2003). This separation of global semantics and local dynamics turns out to be crucial to the success of LTCM.

Recent advances in neural variational inference (Mnih & Gregor, 2014; Miao et al., 2016) have sparked a series of latent variable models applied to conversational modeling (Serban et al., 2016b; Cao & Clark, 2017; Zhao et al., 2017). The majority of the work passes a Gaussian random variable to the hidden state of the LSTM decoder and employs the reparameterisation trick (Kingma & Welling, 2014) to build an unbiased and low-variance gradient estimator for updating the model parameters. However, studies have shown that training this type of models for language generation tasks is tough because the effect of the latent variable tends to vanish and the language model would take over the entire generation process over time (Bowman et al., 2015). This results in several workarounds such as KL annealing (Bowman et al., 2015; Cao & Clark, 2017), word dropout and historyless decoding (Bowman et al., 2015), as well as auxiliary bag-of-word signals (Zhao et al., 2017). Unlike previous approaches, LTCM is similar to TopicRNN (Dieng et al., 2017) where it passes the latent variable to the output layer of the decoder and only back-propagates the gradient of the topic words to the latent variable.

In summary, the contribution of this paper is two-fold: first and most importantly, we show that LTCM can learn to generate more diverse and interesting responses by sampling from the learnt topic representations. The results were confirmed by a corpus-based evaluation and a human assessment; secondly, we conducted a series of experiments to understand the properties of seq2seq-based latent variables models better, which may serve as rules of thumb for future model development.

## 2 BACKGROUND

We present the necessary building blocks of the LTCM model. We first introduce the seq2seq-based conversational model and its latent variable variant, followed by an introduction of the neural topic models.

### 2.1 SEQ2SEQ CONVERSATIONAL MODEL

In general, a seq2seq model (Sutskever et al., 2014) generates a target sequence given a source sequence using two Recurrent Network Networks (RNNs), one for encoding the source, another for decoding the target. Given a *user* input $u = \{x_1, x_2, ...x_U\}$ in the conversational setting, the goal is to produce a *machine* response $m = \{y_1, y_2, ...y_M\}$ that maximises the conditional probability $m^* = \text{argmax}_m \, p(m|u)$. The decoder of the seq2seq model is effectively an RNN language model which measures the likelihood of a sequence through a joint probability distribution,

$$p(m|u) = p(y_1|u) \prod_{t=2}^{M} p(y_t|y_{1:t-1}, u) \tag{1}$$

The conditional probability is then modeled by an RNN,

$$p(y_t|y_{1:t-1}, u) \triangleq p(y_t|\mathbf{h}_t) \tag{2}$$

$$\mathbf{h}_t = f_{\mathbf{W}_h}(y_{t-1}, \mathbf{h}_{t-1}) \tag{3}$$

where $\mathbf{h}_t$ is the hidden state at step $t$ and function $f_{\mathbf{W}_h}(\cdot)$ is the hidden state update that can either be a vanilla RNN cell or a more complex cell like Long Short-term Memory (LSTM) (Hochreiter & Schmidhuber, 1997). The initial state of the decoder $\mathbf{h}_0$ is initialised by a vector representation of the source sentence, which is taken from the last hidden state of the encoder $\mathbf{h}_0 = \hat{\mathbf{h}}_U$. The encoder state update also follows Equation 3.

While theoretically, RNN-based models can memorise arbitrarily long sequences if provided with sufficient capacity, in practice even the improved version such as LSTM or GRU (Chung et al., 2014) encounter difficulties during optimisation (Bengio et al., 1994). This inability to memorising long-term dependencies prevents the model from extracting useful sentence-level semantics. As a consequence, the model tends to focus on the low-hanging fruit (language modelling) during optimisation and yields a suboptimal result.

### 2.2 LATENT VARIABLE CONVERSATIONAL MODELS

Latent variable conversational model (Serban et al., 2016b; Cao & Clark, 2017; Zhao et al., 2017) is a derivative of the seq2seq model in which it incorporates a latent variable $\nu$ at the sentence-level to

inject stochasticity and diversity. The objective function of the latent variable model is

$$p(m|u) = \int_{\nu} p(m|\nu, u)p(\nu|u)d\nu \tag{4}$$

where $\nu$ is usually chosen to be Gaussian distributed and passed to the decoder at every time step where we rewrite Equation 3 as $\mathbf{h}_t = f_{\mathbf{W}_h}(y_{t-1}, \mathbf{h}_{t-1}, \nu)$. Since the optimisation against Equation 4 is intractable, we apply variational inference and alternatively optimise the variational lowerbound,

$$\log p(m|u) = \log \int_{\nu} p(m|\nu, u)p(\nu|u)d\nu$$
$$\geq \mathbb{E}_{q(\nu|u,m)}[\log p(m|\nu, u)] - D_{KL}(q(\nu|u, m)||p(\nu|u)) \tag{5}$$

where we introduce the inference network $q(\nu|u, m)$, a surrogate of $p(\nu|u)$, to approximate the true posterior during training. Based on Equation 5, we can then sample $\nu \sim q(\nu|u, m)$ and apply the Gaussian reparameterisation trick (Kingma & Welling, 2014) to calculate the gradients and update the parameters.

Although latent variable conversational models were able to generate diverse responses, its optimisation has been proven difficult, and several tricks are needed to obtain a good result. Among these tricks, KL loss annealing is the most general and effective approach (Bowman et al., 2015). The main idea of KL annealing is, instead of optimising the full KL term during training, we gradually increase using a linear schedule. This way, the model is encouraged to encode information cheaply in $\nu$ without paying huge KL penalty in the early stage of training.

## 2.3 NEURAL TOPIC MODELS

Probabilistic topic models are a family of models that are used to capture the global semantics of a document set (Srivastava & Sahami, 2009). They can be used as a tool to organise, summarise, and navigate document collections. As an unsupervised approach, topic models rely on counting word co-occurrence in the same document to group words into topics. Therefore, each topic represents a word cluster which puts most of its mass (weight) on this subset of the vocabulary. Despite there are many probabilistic graphical topic models (Blei et al., 2003), we focus on neural topic models (Larochelle & Lauly, 2012; Miao et al., 2016) because they can be directly integrated into seq2seq model as a submodule of LTCM.

One neural topic model that is similar to LDA is the Gaussian-softmax neural topic model introduced by Miao et al. 2017. The generation process works as following:

1. Draw a document-level latent vector $\nu \sim N(\mu_0, {\sigma_0}^2)$.
2. Construct a document-level topic proportion vector $\theta = \text{softmax}(\mathbf{W}^\top \nu)$.
3. For each word $y_t$ in the document,
   (a) Draw a topic assignment $z_t \sim \text{Multinomial}(\theta)$.
   (b) Draw a word $y_t \sim \text{Multinomial}(\beta_{z_t})$.

where $\beta = \{\beta_1, \beta_2, ...\beta_K\}$, $\beta_k$ is the word distribution of topic $k$, and $\mu_0$ and $\sigma_0$ are the mean and variance of an isotropic Gaussian. The likelihood of a document $d = \{y_1, y_2, ...y_D\}$ is therefore,

$$p(d) = \int_{\theta} p(\theta) \prod_{t=1}^{D} \sum_z p(z_t|\theta)p(y_t|\beta_{z_t})d\theta = \int_{\theta} p(\theta) \prod_{t=1}^{D} (\theta\beta)_{y_t} d\theta \tag{6}$$

Note that in the original LDA, both the $\theta$ and $\beta$ are drawn from a Dirichlet prior. Gaussian-softmax model, on the other hand, constrcuts $\theta$ from a draw of an isotropic Gaussian with parameters $\mu_0$ and $\sigma_0$, where as $\beta$ is random initialised as a parameter of the network.

Like most of the topic models, Gaussian-softmax model makes the bag-of-words assumption where the word order is ignored. This simple assumption sacrifices the ability to model local transitions between words and phrases in exchange for the capability to capture global semantics. Therefore, although topic model could not be used as a conversational model itself, it is nevertheless a perfect fit to a sentence-level semantic extractor alongside a seq2seq model to improve the global coherence of the generated responses.

## 3  Latent Topic Conversational Model

**Model**  The proposed Latent Topic Conversational Model (LTCM) is a hybrid of the seq2seq conversational model and the neural topic model, as shown in Figure 1. The neural topic sub-component is responsible for extracting and mapping between the input and output global semantics so that the seq2seq submodule can focus on perfecting local dynamics of the sentence such as syntax and word order. Given a user input $u$ and a machine response $m$, the generative process of LTCM can be described as the following,

1. Encode user prompt u into a vector representation $\mathbf{u} = g_\Gamma(u) \in \mathbb{R}^d$.

2. Draw a sentence-level latent vector $\nu \sim p_\Lambda(\nu|\mathbf{u})$.

3. Construct a sentence-level topic proportion vector $\theta = \text{softmax}(\mathbf{W}_1^\top \nu) \in \mathbb{R}^K$.

4. Initialise the decoder hidden state $\mathbf{h}_0 = \hat{\mathbf{h}}_U$, where $\hat{\mathbf{h}}_U$ is the last encoder state.

5. Given $y_{1:t-1}$, for the $t$-th word $y_t$ in the response,

   (a) Update decoder hidden state $\mathbf{h}_t = f_{\mathbf{W}_h}(y_{t-1}, \mathbf{h}_{t-1})$

   (b) Draw a topic word indicator $l_t \sim \text{Bernoulli}(\text{sigmoid}(\mathbf{W}_2^\top \mathbf{h}_t))$

   (c) Draw a word $y_t \sim p(y_t|\mathbf{h}_t, l_t, \theta; \beta)$, where

$$p(y_t = i|\mathbf{h}_t, l_t, \theta; \beta) \propto \exp(\mathbf{v}_i^\top \mathbf{h}_t + l_t \cdot \beta_i^\top \theta)$$

where $p(\nu|\mathbf{u}) = N(\mu(\mathbf{u}), \sigma^2(\mathbf{u}))$ is a parametric isotropic Gaussian with a mean and variance both condition on the input prompt $\mu(\mathbf{u}) = \text{MLP}(\mathbf{u})$, $\sigma(\mathbf{u}) = \text{MLP}(\mathbf{u})$. To combine the seq2seq model with the neural topic module, we adopt the hard-decision style from TopicRNN (Dieng et al., 2017) by introducing an additional random variable $l_t$. The topic indicator $l_t$ is to decide whether or not to take the logits of the neural topic module into account. If $l_t = 0$, which indicates that $y_t$ is a stop-word, the topic vector $\theta$ would have no contribution to the final output. However, if $l_t = 1$, then the topic contribution term $\beta_i^\top \theta$ is added to the output of the seq2seq model, where $\beta_i$ is the word-topic vector for the $i$-th vocabulary word.

Although the topic word indicator $l_t$ is sampled during inference, during training it is treated as observed and can be produced by either a stop-word list or ranking words in the vocabulary by their inverse document frequencies. This hard decision of $l_t$ is crucial for LTCM because it explicitly sets two gradient routes for the model: when $l_t = 1$ the gradients are back-propagated to the entire network; otherwise, they only flow through the seq2seq model. This is important because topic models are known to be bad at dealing with stop-words (Mimno et al., 2017). Therefore, preventing the topic model to learn from stop-words can help the extraction of global semantics. Finally, the logits of the seq2seq and neural topic model are combined through an additive procedure. This makes the gradient flow more straightforward and the training of LTCM becomes easier[1].

The parameters of LTCM can be denoted as $\Theta = \{\Gamma, \Lambda, \mathbf{W}_1, \mathbf{W}_2, \mathbf{W}_h, \mathbf{V}, \beta\}$ where $\mathbf{V} = \{\mathbf{v}_1, \mathbf{v}_2, ... \mathbf{v}_L\}$ and $L$ is the vocabulary size. During training, the observed variables are input $u$, output $m$, and the topic word indicators $l_{1:M}$. The parametric form of LTCM is therefore,

$$p(m, l_{1:M}|u) = \int_\theta p(\theta|u)p(y_{1:M}, l_{1:M}|\theta, u)d\theta = \int_\theta p(\theta|u) \prod_{t=1}^{M} p(y_t|\mathbf{h}_t, l_t, \theta; \beta)p(l_t|\mathbf{h}_t)d\theta \quad (7)$$

**Inference**  As a direct optimisation of Equation 7 is intractable because it involves an integral over the continuous latent space, variational inference (Jordan et al., 1999) is applied to approximate the log-likelihood objective. The variational lowerbound of Equation 7 can therefore be derived as

$$\mathcal{L} = \mathbb{E}_{q(\theta|u,m)}\left[\sum_{t=1}^{M} \log p(y_t|\mathbf{h}_t, l_t, \theta; \beta) + \sum_{t=1}^{M} \log p(l_t|\mathbf{h}_t)\right] - D_{KL}(q(\theta|u,m)||p(\theta|u))$$

$$\leq \int_\theta p(\theta|u) \prod_{t=1}^{M} p(y_t|\mathbf{h}_t, l_t, \theta; \beta)p(l_t|\mathbf{h}_t)d\theta \quad (8)$$

---

[1]For example, LTCM does not need to be trained with KL annealing to achieve a good performance.

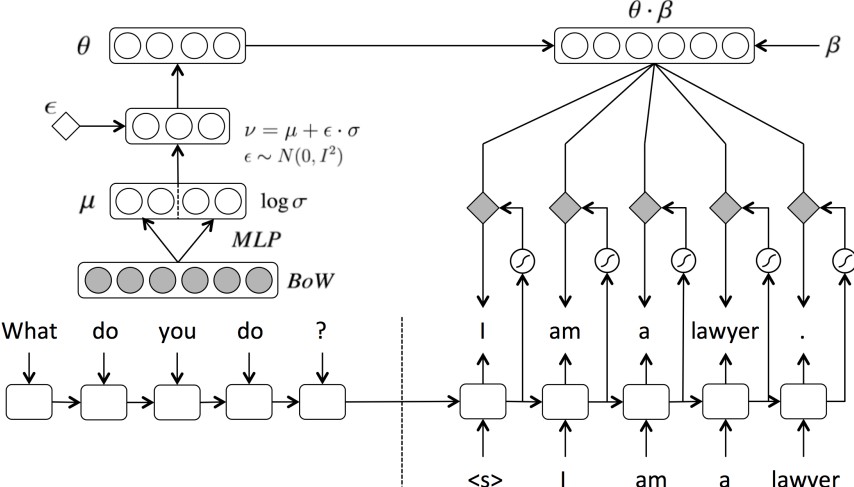

Figure 1: The graphical representation of the Latent Topic Conversational Model. The shaded nodes are observed while the blank nodes are hidden. The circles are neurons in neural network layers, rectangles are the LSTM cells, and the diamonds are stochastic nodes.

where $q(\theta|u, m)$ is the inference network introduced during training to approximate the true posterior. The neural variational inference framework (Mnih & Gregor, 2014; Miao et al., 2016) and the Gaussian reparameterisation trick (Kingma & Welling, 2014; Rezende et al., 2014) are then followed to construct $q(\theta|u, m)$,

$$q(\theta|u, m) = \text{softmax}(\mathbf{W}_a^\top \nu'), \nu' \sim N(\nu|\mu(u, m), \sigma^2(u, m)) \tag{9}$$

$$\mu(u, m) = \text{MLP}_{\Omega_1}(u_b, m_b), \sigma(u, m) = \text{MLP}_{\Omega_2}(u_b, m_b) \tag{10}$$

where $\Phi = \{\mathbf{W}_a, \Omega_1, \Omega_2\}$ is the new set of parameters introduced for the inference network, $u_b$ and $m_b$ are the bag-of-words representations for $u$ and $m$, respectively. Although $q(\theta|u, m)$ and $p(\theta|u)$ are both parameterised as an isotropic Gaussian distribution, the approximation $q(\theta|u, m)$ only functions during training by producing samples to compute the stochastic gradients, while $p(\theta|u)$ is the generative distribution that generates the required topic proportion vectors for composing the machine response.

# 4 EXPERIMENTS

**Dataset**  We assessed the performance of the LTCM using both a corpus-based evaluation and a human assessment. The dataset used in the experiments is a subset of the data collected by Shao et al. 2017, which includes mainly the Reddit[2] data which contains about 1.7 billion messages (221 million conversations). Given the large volume of the data, a random subset of 15 million single-turn conversations was selected for this experiment. To process the Reddit data, messages belonging to the same post are organized as a tree, a single-turn conversation is extracted merely by treating each parent node as a prompt and its corresponding child nodes as responses. A length of 50 words was set for both the source and target sequences during preprocessing. Sentences with any non-Roman alphabet were also removed. This filters out around 40% to 50% of the examples. A few standardizations were made via regular expressions such as mapping all valid numbers to <number> and web URLs to <url>. A vocabulary size of 30K was set for encoder, decoder, and the neural topic component.

**Model**  The LTCM model was implemented on the publicly available NMT[3] code base (Luong et al., 2017). Three model types were compared in the experiments, the vanilla seq2seq conversational model (S2S) (Vinyals & Le, 2015), the latent variable conversational model (LV-S2S) (Serban

---

[2]Available at https://goo.gl/9gKEbc.

[3]Available at https://github.com/tensorflow/nmt

| Model | ppx | lowerbound | kl | unique(%) | zipf |
|---|---|---|---|---|---|
| S2S; greedy | 46.26 | 69.20 | 0.00 | 2.65 | 1.14 |
| S2S; sample | | | | 96.73 | 1.07 |
| LV-S2S, $p(\nu)$ | 46.10 | 69.14 | 39.11 | 3.27 | 1.14 |
| LV-S2S, $p(\nu|u)$ | 45.99 | 69.10 | 39.09 | 3.07 | 1.14 |
| LV-S2S, $p(\nu|u)$, +A | 47.54 | 78.01 | 47.74 | 42.62 | 1.13 |
| LTCM, $p(\theta)$ | 95.19 | 91.18 | 55.47 | 50.34 | 1.11 |
| LTCM, $p(\theta|u)$ | **45.24** | 89.17 | 59.29 | **54.08** | 1.11 |
| LTCM, $p(\theta|u)$, +V | 45.47 | 85.89 | 55.97 | 48.83 | 1.12 |

Table 1: Result of the corpus-based evaluation. $p(\nu)$ or $p(\theta)$ means the model samples from a gaussian prior, while $p(\nu|u)$ or $p(\theta|u)$ means the model samples from a Gaussian conditional distribution. +A indicates the model is trained with KL annealing, while +V means the model has a larger stop-word vocabulary (500).

et al., 2016b; Cao & Clark, 2017), and the Latent Topic Conversational Model (LTCM). For all the seq2seq components, a 4-layer LSTM with 500 hidden units was used for both the encoder and decoder. We used the GNMT style encoder (Wu et al., 2016) where the first layer is a bidirectional LSTM, while the last three layers are unidirectional. Residual connections were used (He et al., 2016) to ease the optimisation of deep networks. Layer Normalisation (Ba et al.) was applied to all the LSTM cells to facilitate learning. The batch size was 128, and a dropout rate of 0.2 was used. The Adam optimiser (Kingma & Ba, 2014) with a fixed annealing schedule was used to update the parameters. For the latent variable conversational model, we explored the KL annealing strategy as suggested in Bowman et al. 2015 where the KL loss is linearly increased and reaches to the full term after one training epoch. In LTCM, the 300 words with the highest inverse document frequency are used as stop-words and the rest are treated as topic words. Both the mutual angular regularisation (Xie et al., 2016) and the l2 regularisation were applied to the $\beta$ matrix during training.

**Evaluation and Decoding**   To build the development and testing sets, additional 20K sentence pairs were extracted and divided evenly. For evaluation, five metrics were reported: the approximated perplexity, the variational lowerbound, the KL loss, the sentence uniqueness and the Zipf coefficient (Cao & Clark, 2017) of the generated responses. Because the exact perplexity of the latent variable models is hard to assess due to sampling, an approximated perplexity is reported as suggested in Dieng et al. 2017. For latent variable conversational models, the approximate distribution for computing perplexity is $p(y_t|y_{1:t-1}, u) = \prod_t p(y_t|\mathbf{h}_t, \hat{\nu})$, where $\hat{\nu}$ is the mean estimate of $\nu$. While for LTCM it is

$$p(y_t|y_{1:t-1}, u) = \prod_t p(y_t|\mathbf{h}_t, l_t, \hat{\theta}; \beta)p(l_t|\mathbf{h}_t) \tag{11}$$

where again $\hat{\theta}$ is the mean estimate of $\theta$. Both latent variable model and LTCM used greedy decoding to make sure the diversity they produce comes from the latent variable. For seq2seq model, however, we explored both the greedy and random sampling strategies. Given a prompt, each model was requested to generate five responses. This leads to 50K generated responses for the testing set. The sentence uniqueness score and Zipf coefficient[4], which were introduced both by Cao & Clark 2017 as proxies to evaluate sentence and lexicon diversity respectively, were computed.

## 4.1 CORPUS-BASED EVALUATION RESULT

The result of the corpus-based evaluation is presented in Table 1. The first block shows the performance of the baseline seq2seq model, either by greedy decoding or random sampling. Unsurprisingly, *S2S-sample* can generate much more diverse responses than *S2S-greedy*. However, these responses are not of high quality as can be seen in the human assessment in the next section. One interesting observation is that the sentence uniqueness score of *S2S-greedy* is much lower than the ex-

---

[4]Note, a higher sentence uniqueness and a lower Zipf coefficient indicates that the result is more diverse.

| Model | S2S, greedy | S2S, sample | LV-S2S, $p(\nu|u)$ +A | LTCM, $p(\theta|u)$ |
|---|---|---|---|---|
| Interestingness | 3.43 (3.43) | 3.80 (3.00) | 3.90 (3.41) | 3.97 (3.37) |
| Appropriateness | 3.53 (3.53) | 3.68 (2.76) | 3.96 (3.41) | 4.04 (3.36) |

Table 2: Quality assessment. Both metrics were rated from 1 to 5. The numbers inside the brackets are computed by averaging the mean of the generated responses across prompts, while the ones outside the brackets are the average of the maximum scores across prompts.

| Preference (%) | S2S, greedy | S2S, sample | LV-S2S, $p(\nu|u)$, +A | LTCM, $p(\theta|u)$ |
|---|---|---|---|---|
| S2S, greedy | - | 48.2 | 39.3 | 33.3 |
| S2S, sample | 51.8 | - | 36.9 | 38.8 |
| LV-S2S, $p(\nu|u)$, +A | 60.7 | 63.1 | - | 40.0 |
| LTCM, $p(\theta|u)$ | 66.7[*] | 61.2[*] | 60.0 | - |

Table 3: Pairwise preference assessment. Note the numbers are the percentage of wins when comparing models in the first column with the ones in first row. *$p < 0.05$

pected (2.65%$<$ 20%[5]). This echoes the generic response problem mentioned in previous works (Li et al., 2016a; Serban et al., 2016b). The second block demonstrates the result of the latent variable conversational models. As can be seen, neither sampling from a prior (*LV-S2S, $p(\nu)$*) nor a conditional (*LV-S2S, $p(\nu|u)$*) helps to beat the performance of the seq2seq model. Although both models perform equally well in terms of perplexity and lowerbound, the likewise low uniqueness scores as seq2seq indicate that both of their latent variables collapse into a single mode and do not encode much information. This was also observed in Zhao et al. 2017 when training seq2seq-based latent variable models. The KL annealed model *LV-S2S, $p(\nu|u)$, +A*, as suggested by Bowman et al. 2015, can help to mitigate this problem and achieve a much higher uniqueness score (42.6%).

The third block shows the result of the LTCM models. As can be seen, LTCM trades in its KL loss and variational lowerbound in exchange for a higher response diversity (higher uniqueness score and lower Zipf). Interestingly, although the lowerbound was substantially worse than the baselines, the conditional LTCM models (*LTCM, $p(\theta|u)$* and *LTCM, $p(\theta|u)$, +V*) can still reach comparable perplexities. This indicates that most of the additional loss incurred by LTCM was to encode the discourse-level diversity into the latent variable and therefore may not be a bad idea. Given that the latent variable of LTCM can encode more useful information, sampling from a conditional can therefore better tailor the neural topic component to the user prompt and produce more relevant responses (*LTCM, $p(\theta)$* v.s. *LTCM, $p(\theta|u)$*). Overall speaking, LTCM can generate more diverse responses comparing to baselines by encoding more information into the latent space. However, the slightly higher lowerbound and KL loss do not necessarily mean that the quality of the responses is worse. More discussions follow in the next section.

## 4.2 HUMAN EVALUATION

Due to the difficulty in evaluating conversational agents (Liu et al., 2016; Wen et al., 2015), a human evaluation is usually necessary to assess the performance of the models. To do a less biased evaluation, a set of judges ($\sim$ 250) were recruited on AMT. For each task (a prompt), two randomly selected models were paired and each of them was asked to generate five responses given the prompt. There is a total of 5000 comparisons randomly split between all pairs. This results in approximately 90 experiments per pair of comparison. The number of tasks that each judge can do is capped to 20. To consider the response diversity, each judge was asked to rate each of the five generated responses from 1 to 5 based on the *interestingness* and *appropriateness* scores. The quality assessment is shown in Table 2. The numbers inside the brackets are calculated by averaging the mean of the generated responses across prompts, while the ones outside the brackets are the average of the maximum scores across prompts. Moreover, at the end of the task, the judge was also asked to state a preference between the two systems. The result is shown in Table 3.

---

[5]A deterministic model which is forced to decode the same prompt five times should ideally reach 20% uniqueness.

| Model | Responses |
|---|---|
| **Prompt**: what do you think about messi ? | |
| **S2S** | * i think he 's a good player . |
| **LV-S2S+A** | * he 's a fantastic player , but he 's not a good player . |
| | * he 's a great player , but he 's not a good player . |
| | * he 's a great player , but he needs to be more consistent . |
| **LTCM** | * i love him . |
| | * i think he 's a good player , but i feel like he 's a bit overrated . |
| | * i think he 's a great player , but i do not think messi deserves to play for the rest of the season . |
| | * i think messi is the best . |
| **Prompt**: what is the purpose of existence ? | |
| **S2S** | * to create a universe that is not a universe . |
| **LV-S2S+A** | * to be able to understand what you are saying . |
| **LTCM** | * to be a \<unk\> . |
| | * to be able to see the world . |
| | * to be able to see things . |
| | * to make it better . |

Table 4: Example comparisons of the three models: *S2S-greedy*, *LV-S2S*, $p(\nu|u)$, *+A*, and *LTCM*, $p(\theta|u)$. The result is produced by removing duplicated sentences from the five generated responses. Each * indicates the beginning of a response.

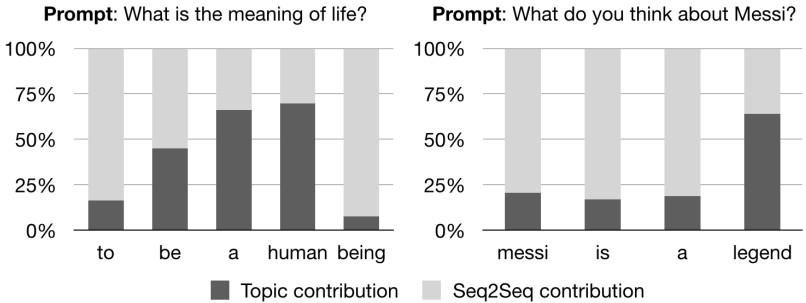

Figure 2: Analysis of the learned topic gate $l_t$ shown in percentage.

Table 2 shows that the average scores (numbers inside the brackets) of *S2S-greedy*, *LV-S2S*, $p(\nu|u)$, *+A*, and *LTCM*, $p(\theta|u)$ are pretty much the same (with the *appropriateness* of *S2S-greedy* slightly better). However, the maximum scores (numbers outside the brackets) show that LTCM is the best among the four (*interestingness*: 3.97 and *appropriateness*: 4.04). This indicates that although LTCM can generate pretty good responses, it could also produce sentences that are quite bad. This variance in response quality could be beneficial if reinforcement learning is introduced to fine-tune the latent variable (Wen et al., 2017a). Table 3 shows the result of pairwise preference test between four models. As can be seen, LTCM is the preferred option for most of the judges when compared to other approaches. Table 4 shows a few examples for qualitative analysis of the models. As shown in the table, LTCM can generate more diverse and interesting responses comparing to the baseline methods. The diversity can be found at both the semantic and the syntactic level. Figure 2 shows the analysis of the topic gate. As can be seen, the learned gate corresponds to the human intuition and helps to coordinate the contribution of the two models while generating. For more examples of the generated responses, please refer to Appendix A.

## 5 CONCLUSION

In this paper, we have proposed the Latent Topic Conversational Model (LTCM) for general-purpose conversational modeling. We have shown that LTCM can generate more interesting and diverse responses by combining the seq2seq model and neural topic model so that global semantic representations and local word transitions can be modeled separately but learned jointly. Both a corpus-based evaluation and a human assessment confirm this finding. Future work would be to study the learned representations and use them to control the meaning of the generated responses.

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

## A   EXAMPLE RESPONSES

| Model | Responses |
|---|---|
| **Prompt**: What is the purpose of dying? | |
| S2S | * Dying |
| LV-S2S+A | * The fact that you are not a doctor . |
| | * The fact that you are not alone is not a good thing . |
| LTCM | * To die . |
| | * To make it a better life . |
| | * To make it a reality . |
| **Prompt**: aerial view of colpitt lake in halifax | |
| S2S | * i 'm going to go out on a limb and say that the lake is a lake . |
| LV-S2S+A | * i bet it 's a lot of fun to drive through there . |
| | * i love this place . . . i 've been there a few times . it 's fucking amazing . |
| | * i was there last year . . . it was really nice . |
| | * i wish i could have been there . i was in the <unk> and it was pretty cool . |
| | * wow , that 's a lot of water . |
| LTCM | * i love this place . |
| | * wow , great shot ! |
| | * wow , that 's beautiful . thanks for the post op . |
| | * wow , that looks amazing . |
| | * wow , that looks amazing . thanks for sharing . |
| **Prompt**: ( spoilers ) what are your season finale predictions ? | |
| S2S | * i 'm going to be a <unk> fan . |
| LV-S2S+A | * i think the <unk> will be the last one . . . |
| | * i think the <unk> will be the next <unk> . |
| | * i think the <unk> will be the one to go . |
| LTCM | * i think it 's a pretty good idea . |
| | * i think it 's going to be a tie between the <unk> and <unk> . |
| | * jon <unk> dies . |
| | * jon snow will be dead . |
| | * jon snow will die . |
| **Prompt**: <number> miles to my local disk golf course . what a beautiful day <smile> | |
| S2S | * i 'm jealous . i 'm jealous . |
| LV-S2S+A | * i 'm jealous . |
| | * i 'm jealous . . . |
| | * i 'm jealous of your disc golf course . i 've been looking for a good one for a while now . |
| | * i 've always wanted to do this . . . but i 'm not sure if i should let it go . |
| | * i really like the color of the disc . |
| LTCM | * i 'm in the middle of nowhere . i 'm in south florida . |
| | * i 'm jealous . i 've never played golf in years . |
| | * i love disc golf . i have a disc golf course in missouri . i 'm going to try to get a disc golf course in the next couple of week . |
| | * nice ! looks great <smile> |
| | * nice ! looks like a great course . |
| **Prompt**: i 've been waiting all year and it 's finally happening | |
| S2S | * i 'm still waiting for the <number> th . |
| LV-S2S+A | * i 'm so excited ! |
| LTCM | * congratulations ! ! ! |
| | * You 're a little late to the party , but you 're still in the process of making a purchase . |
| | * you 're a little late to the party , but you 're still young . |
| | * you 're a lucky man . congrats ! |
| | * you 're lucky you got it . congrats ! |

Table 5: Extended examples of Table 4. The result is produced by removing duplicated sentences from the five generated responses. Each * indicates the beginning of a response.

| Model | Responses |
|---|---|
| **Prompt**: What is your job ? | |
| S2S | * i'm a <unk> . |
| LV-S2S+A | * i'm a <unk> . |
| LTCM | * i'm a software engineer . i'm working on building a new computer . |
| | * i'm a software engineer . i spend most of my summer time working on my computer ... |
| | * i enjoy working at mcdonalds . |
| | * working on a project management company . |
| **Prompt**: If you can change one thing about your country, what would you change ? | |
| S2S | * i'd be happy to be a dictator . |
| LV-S2S+A | * i'd make a lot of money and make money off of it . |
| LTCM | * i would not be able to change my country . |
| | * i would not be able to change my mind . |
| | * i would vote republican . |
| | * mandatory healthcare reform . |
| **Prompt**: I want a poster of this ! | |
| S2S | * i want one . |
| LV-S2S+A | * i want one . |
| | * i want one ! |
| LTCM | * i want one ! |
| | * i approve . |
| **Prompt**: what movie do you find hilarious but no one else does ? | |
| S2S | * the big lebowski . |
| LV-S2S+A | * the <unk> . |
| | * the hangover |
| | * the hangover . it's really funny . |
| LTCM | * anything involving nicolas cage . |
| | * anything with a twist . |
| | * anything with nicholas cage . |
| | * the godfather . |
| **Prompt**: what kind of operating system you are using ? | |
| S2S | * <unk> |
| LV-S2S+A | * <unk> . |
| | * ubuntu . |
| LTCM | * i'm a <unk> . |
| | * i'm a linux user . |
| | * i use <unk> <smile> |
| | * linux . |
| | * ubuntu linux . |
| **Prompt**: i created the universe ! give me the gift certificate ! | |
| S2S | * i'm not sure if i should be happy or sad . |
| LV-S2S+A | * i always get really excited when i see this . |
| | * i hope he's okay . |
| | * i need to get a job . |
| | * i really like this . thanks for sharing . |
| | * i want to see this . |
| LTCM | * i love this . |
| | * oh god , i love this guy . |
| | * oh my god , i love this . |
| | * oh my god , i love this guy . |
| | * oh my god , yes ! |

Table 6: Extended examples of Table 4. The result is produced by removing duplicated sentences from the five generated responses. Each * indicates the beginning of a response.

