# OpenReview forum: "Latent Topic Conversational Models"
_ICLR.cc/2018/Conference — Reject_

### Official Review · AnonReviewer2 · 2017-11-28
**The evaluation and details of experiments are not sufficient.**

**Rating:** 4
**Confidence:** 3

**Review:**

This paper proposed the combination of topic model and seq2seq conversational model.
The idea of this combination is not surprising but the attendee of ICLR might be interested in the empirical results if the model clearly outperforms the existing method in the experimental results.
However, I'm not sure that the empirical evaluation shows the really impressive results.
In particular, the difference between LV-S2S and LTCM seem to be trivial.
There are many configurations in the LSTM-based model.
Can you say that there is no configuration of LV-S2S that outperforms your model?
Moreover, the details of human evaluation are not clear, e.g., the number of users and the meaning of each rating.

---

> ### Public Comment · (anonymous) · 2018-01-03
> *** Responses to R2**
>
> Hyperparameter search of LV-S2S
> The experiments were conducted in a careful way where a small set of hyper-parameters were tuned to find the best model in each category. We didn’t do an exhaustive grid search over all possible network configurations, however, given the recent understanding of latent variable models (Higgins et al, 2016, Bowman et al., 2015, Dieng et al., 2017), the result of this work has shown good evidences that LTCM is generally more capable of learning diverse and interesting responses than latent variable S2S models.
>
> Human evaluation
> We ran 5000 pairwise comparisons between the 8 models in Table 1 (~90 comparisons per pair) and reported only the top performing ones in each of the model categories. The number of tasks each MTurk can work on was capped at 20. This results in about >=250 unique workers. The meaning of each rating is presented in Section 4.2 Human evaluation. We have added these details, please see our revision.

---

### Official Review · AnonReviewer1 · 2017-11-28
**interesting combination of seq2seq and neural topic models, but weak evaluation**

**Rating:** 5
**Confidence:** 4

**Review:**

The paper proposes a conversational model with topical information, by combining seq2seq model with neural topic models. The experiments and human evaluation show the model outperform some the baseline model seq2seq and the other latent variable model variant of seq2seq.

The paper is interesting, but it also has certain limitations:

1) To my understanding, it is a straightforward combination of seq2seq and one of the neural topic models without any justification.
2) The evaluation doesn't show how the topic information could influence word generation. No of the metrics in table 2 could be used to justify the effect of topical information.
3) There is no analysis about the model behavior, therefore there is no way we could get a sense about how the model actually works. One possible analysis is to investigate the values $l_t$ and the corresponding words, which to some extent will tell us how the topical information be used in generation. In addition, it could be even better if there are some analysis about topics extracted by this model.

This paper also doesn't pay much attention to the existing work on topic-driven conversational modeling. For example "Topic Aware Neural Response Generation" from Xing et al., 2017.

Some additional issues:

1) In the second line under equation 4, y_{t-1} -> y_{t}
2) In the first paragraph of section 3, two "MLP"'s are confusing
3) In the first paragraph of page 6, words with "highest inverse document frequency" are used as stop words?

---

> ### Public Comment · (anonymous) · 2018-01-03
> *** Responses to R1**
>
> Literature survey
> The authors have updated the paper per the reviewer’s suggestion to add more citations for topic-aware models. We would like to point out that the suggested work (Xing et al., 2017) is quite different from ours in that they used a pretrained LDA model whereas our LTCM model trains the topic and seq2seq component jointly.
> [1] Xing et al., 2017. Topic Aware Neural Response Generation. https://arxiv.org/pdf/1606.08340.pdf
>
> Interpret topical information
> The topic information learned in LTCM was not as easy to interpret as in other topic models that trained on document sets. This is because the word co-occurrence statistics in short text datasets are too sparse to train interpretable topic representations (Yan et al, 2013). However, we found that sampling from this learned latent representation does give us diversified sentences, at both syntactic and semantic levels. We do acknowledge the suggestion to visualize the values of $l_t$ which we have included in the newest revision of the paper.

---

### Official Review · AnonReviewer3 · 2017-12-02
**topic modeling + seq2seq**

**Rating:** 6
**Confidence:** 4

**Review:**

I enjoyed this paper a lot. The paper addresses the issue of enduring topicality in conversation models. The model proposed here is basically a mash-up between a neural topic model and a seq2seq-based dialog system. The exposition is relatively clear and a reader with sufficient background in ML should have no following the model. My only concern about the paper is that is very incremental in nature -- the authors combine two separate models into a relatively straight-forward way. The results do are good and validate the approach, but the paper has little to offer beyond that.

---

> ### Public Comment · (anonymous) · 2018-01-03
> *** Responses to R3**
>
> We thank the reviewer for the comments. Please see this post (– General comments on Contributions –) for the contributions of our paper.

---

### Public Comment · (anonymous) · 2018-01-03
**– General comments on Contributions –**

We thank all the reviewers for the comments and feedback, which have helped us improve the paper (please see our revision). However, we are disappointed about the low review scores of the paper and that our contributions were not fully appreciated. To help reviewers better evaluate our paper, we would like to re-emphasize the contributions of this work:

(a) Novelty
We were first to be able to jointly learn the neural topic and seq2seq models. The key idea is to utilize the hard-decision trick from TopicRNN (Dieng et al., 2017) to prevent the latent variable from catastrophic mode collapsing. Previous work such as [1, 2] only incorporated pre-trained models (LDA, counting grid) into seq2seq models instead of joint learning.
[1] Xing et al., 2017. Topic Aware Neural Response Generation. https://arxiv.org/pdf/1606.08340.pdf
[2] Wang et al., 2017. Steering Output Style and Topic in Neural Response Generation.
https://arxiv.org/abs/1709.03010.pdf

(b) Better understanding/training of latent models for languages
Latent models for languages are notoriously hard to train [3, 4]. This work contributes to better training/understanding of latent models by observing and investigating in correlations of many training metrics. For examples, we found that:
  (i)  approximated perplexity has much more to do with the generation quality comparing to variational lower bound;
  (ii) a lower lowerbound isn’t necessarily better because the higher KL can lead to a higher sentence diversity.
  (iii) BoW encoder works just fine in the topic component of LTCM. It is also easier to optimise.
These could serve as valuable rules of thumb for future model development.

[3] Bowman et al., 2015. Generating Sentences from a Continuous Space. https://arxiv.org/pdf/1511.06349.pdf
[4] Miao and Blunsom, 2016. Language as a Latent Variable: Discrete Generative Models for Sentence Compression. https://arxiv.org/pdf/1609.07317.pdf

(c) Standard and comprehensive evaluation
We acknowledge that evaluating chat-based systems is hard. To our best effort, we included previous metrics [5, 6] to provide a comprehensive and extensive evaluation that demonstrates the superiority of our models over strong baselines. The evaluation includes both corpus-based metrics (perplexity, lowerbound, KL divergence, uniqueness, Zipf coefficients) and human judgments (interestingness, appropriateness, as well as a pairwise comparison).

[5] Serban et al., 2016. A Hierarchical Latent Variable Encoder-Decoder Model for Generating Dialogues. https://arxiv.org/pdf/1605.06069.pdf
[6] Cao & Clark, 2017. Latent Variable Dialogue Models and their Diversity. https://arxiv.org/pdf/1702.05962.pdf

---

### Author Response · Authors · 2018-01-05
**Notification of the newest paper update.**

The authors would like to notify reviewers about the newest update of the paper where a quick analysis of the learned topic gate $l_t$ has been added to the paper based on reviewer1's request.

---

### Public Comment · (anonymous) · 2018-01-15
**A combination of VAE and TopicRNN?**

It seems that the model is a combination of VAE (https://arxiv.org/abs/1605.06069) and TopicRNN (https://arxiv.org/pdf/1611.01702.pdf). Any new insights?

---

> ### Author Response · Authors · 2018-01-21
> **Contribution/Novelty of the paper**
>
> The main contribution of the paper is three-fold as mentioned in this post (– General comments on Contributions–):
> 1) We were first to be able to jointly learn the neural topic and seq2seq models.
> 2) The paper offers a better understanding/training of latent models for languages.
> 3) Both an extensive evaluation and a comprehensive analysis were conducted to validate the results.

---

### Decision · Program_Chairs · 2018-01-29
**ICLR 2018 Conference Acceptance Decision**

**Decision:**

Reject

**Comment:**

This paper combines existing models to detect topics and generate responses, and the resulting model is shown to be slightly preferred by human evaluators over baselines. This is quite incremental and the results are not impressive enough to stand on their own merit.